# Variance Matters: Improving Domain Adaptation via Stratified Sampling

**Andrea Napoli & Paul White**                    *{A.Napoli,P.R.White}@soton.ac.uk*
*Institute of Sound and Vibration Research*
*University of Southampton, UK*

**Reviewed on OpenReview:** *https://openreview.net/forum?id=MVwgedTIUs*

## Abstract

Domain shift remains a key challenge in deploying machine learning models to the real world. Unsupervised domain adaptation (UDA) aims to address this by minimising domain discrepancy during training, but the discrepancy estimates suffer from high variance in stochastic settings, which can stifle the theoretical benefits of the method. This paper proposes Variance-Reduced Domain Adaptation via Stratified Sampling (VaRDASS), the first specialised stochastic variance reduction technique for UDA. We consider two specific discrepancy measures – correlation alignment and the maximum mean discrepancy (MMD) – and derive ad hoc stratification objectives for these terms. We then present expected and worst-case error bounds, and prove that our proposed objective for the MMD is theoretically optimal (i.e., minimises the variance) under certain assumptions. Finally, a practical k-means style optimisation algorithm is introduced and analysed. Experiments on four domain shift datasets demonstrate improved discrepancy estimation accuracy and target domain performance.

## 1 Introduction

Machine learning models often underperform when the test data distribution differs from the training distribution, a phenomenon known as domain shift. Improving robustness to domain shift has been a longstanding goal in machine learning, and is crucial to the widespread deployment of AI (Gulrajani & Lopez-Paz, 2021; Koh et al., 2021).

Unsupervised domain adaptation (UDA) is a common approach, where models learn representations invariant across source and target domains by minimising a "domain discrepancy" loss. Two widely used objectives are the correlation alignment (CORAL) loss, which matches covariance matrices (Sun & Saenko, 2016), and the maximum mean discrepancy (MMD), which aligns kernel mean embeddings (Tzeng et al., 2014; Long et al., 2015; Li et al., 2018). Although theoretically well-grounded (Ben-David et al., 2006; 2010; Redko et al., 2022), a key limitation is that empirically estimating these discrepancies is extremely noisy, especially in high dimensions and with small minibatches. This can destabilise training, and sometimes even lead to worse target domain performance than with no domain alignment at all (Dubey et al., 2021; Gao et al., 2023; Gulrajani & Lopez-Paz, 2021; Koh et al., 2021; Napoli & White, 2023; 2024b; Wang et al., 2019).

Stochastic variance reduction offers a natural remedy. However, many classical such methods, including SAGA (Defazio et al., 2014), SVRG (Johnson & Zhang, 2013), and dual coordinate ascent (Shalev-Shwartz & Zhang, 2013), are incompatible with MMD and CORAL, since these losses lack finite-sum structure. Momentum-based approaches (Polyak, 1964; Polyak & Juditsky, 1992; Kingma & Ba, 2014) avoid this problem, but at the cost of biasing the gradients, since the training examples are not weighted equally (Gower et al., 2020). Estimator shrinkage (Ledoit & Wolf, 2020; Muandet et al., 2016; Danafar et al., 2014) may also be applied, but again this introduces bias.

A more flexible direction is to alter the sampling distribution, then correct the sampling bias using weighted losses. Importance sampling biases selection towards points with higher gradient impact (Alain et al., 2015; Johnson & Guestrin, 2018; Katharopoulos & Fleuret, 2017; 2018; Kutsuna, 2025; Loshchilov & Hutter, 2016; Zhao & Zhang, 2015). Similarly, typicality sampling upweights examples in higher-density regions (Peng et al., 2020). Again, these approaches require access to per-sample gradients, and are thus incompatible with our problem. Diverse sampling methods reduce variance by enforcing dissimilarity within minibatches, for example using antithetic pairs (Liu & Xu, 2018), repulsive point processes (Zhang et al., 2017; 2019; Bardenet et al., 2021; Napoli & White, 2024b), or faster heuristics such as k-means++ (Napoli & White, 2024b); such methods are also helpful for balancing the data Zhang et al. (2017); Napoli & White (2024a).

Finally, stratified sampling partitions the data into strata and samples independently from each. This can be considered a special case of diverse sampling (Zhang et al., 2017), but is more computationally efficient since the only significant cost is in stratifying the data, rather than drawing the samples themselves. Zhao & Zhang (2014) cluster the input space directly, which offers an iteration-independent stratification, but this assumes the gradients are Lipschitz continuous with respect to the inputs. Alternatively, Liu et al. (2020b) reconstruct the clusters periodically during training, and propose a criterion to determine when to do so. Fu & Zhang (2017) and Lu et al. (2021) extend stratified sampling to Bayesian and time-series learning respectively.

This paper proposes Variance-Reduced Domain Adaptation via Stratified Sampling (VaRDASS), the first specialised variance reduction strategy for UDA. By using stratified data samplers, VaRDASS obtains variance-reduced stochastic losses without having to trade off bias or make any other compromises with the model or learning algorithm. Moreover, we derive specific stratification objectives for two widely-used UDA losses – MMD and CORAL, present expected and worst-case error bounds, and prove that our proposed objective for the MMD is theoretically optimal (i.e., minimises the variance) under certain assumptions. We then propose a practical k-means style alternating optimisation algorithm to solve this problem in tractable time.

In experiments and simulations, we validate our choices of clustering objectives, whilst showing that VaR-DASS both reduces the estimation error of the domain discrepancy for a given minibatch size, and improves target domain accuracy on 4 UDA benchmark datasets.

## 2 Method

### 2.1 Preliminaries

In UDA, we are given labelled examples $\mathcal{D}_s = \{(x_{s,i}, y_{s,i})\}_{i=1}^{n_s}$ from a source distribution $P_s$, and unlabelled examples $\mathcal{D}_t = \{x_{t,j}\}_{j=1}^{n_t}$ from a different target distribution $P_t$. The goal is to produce a model $\Theta : \mathcal{X} \to \mathcal{Y}$ such that the target risk $\mathbb{E}_{(x_t, y_t) \sim P_t} [L_{\text{task}} (\Theta (x_t), y_t)]$ for some task loss $L_{\text{task}}$ is minimised. We assume $\Theta$ decomposes into a featuriser $\Theta_F : \mathcal{X} \to \mathcal{Z}$ and prediction head $\Theta_C : \mathcal{Z} \to \mathcal{Y}$, and has intermediate feature representations $z_s, z_t$. Since the target data are unlabelled, UDA methods instead minimise $L_{\text{task}}$ on the source domain, alongside a domain adaptation term $L_{\text{DA}}$ which aligns the source and target feature distributions:

$$\min_{\Theta} \mathbb{E}_{(x_s, y_s) \sim P_s, \ x_t \sim P_t} [L_{\text{task}} (\Theta (x_s), y_s) + \lambda \ L_{\text{DA}} (z_s, z_t)], \tag{1}$$

where $\lambda \in \mathbb{R}^+$ is a trade-off parameter. During training, minibatches $\widetilde{\mathcal{D}}_s = \{(\widetilde{x}_{s,h}, \widetilde{y}_{s,h})\}_{h=1}^{k} \subset \mathcal{D}_s$ and $\widetilde{\mathcal{D}}_t = \{\widetilde{x}_{t,h}\}_{h=1}^{k} \subset \mathcal{D}_t$ are randomly sampled at each iteration (we assume for simplicity that $|\widetilde{\mathcal{D}}_s| = |\widetilde{\mathcal{D}}_t| = k$), and used to compute stochastic losses $\widehat{L}_{\text{task}} (\Theta (\widetilde{x}_s), \widetilde{y}_s)$ and $\widehat{L}_{\text{DA}} (\widetilde{z}_s, \widetilde{z}_t)$.

Our aim is to reduce $\text{Var} \left( \widehat{L}_{\text{DA}} \right)$ by constructing the minibatches using stratified sampling. In the following sections, we will formulate and justify stratification objectives to this end for the MMD and CORAL losses, and propose a practical clustering algorithm to solve this problem. Since $\widetilde{\mathcal{D}}_s$ and $\widetilde{\mathcal{D}}_t$ are sampled independently, they can also be stratified independently. The following sections describe the process for $\mathcal{D}_s$; the same procedure can be analogously repeated for $\mathcal{D}_t$.

## 2.2 Optimal stratification for MMD

The (squared) MMD loss is

$$L_{\text{MMD}} = \|\mu_{\phi,s} - \mu_{\phi,t}\|_{\mathcal{H}}^2 \tag{2}$$

where $\mu_{\phi,s} = \mathbb{E}\left[\phi\left(z_s\right)\right]$, $\mu_{\phi,t} = \mathbb{E}\left[\phi\left(z_t\right)\right]$, $\mathcal{H}$ is a reproducing kernel Hilbert space, and $\phi : \mathcal{Z} \to \mathcal{H}$ is an implicit mapping. $\mathcal{H}$ is associated with a unique positive-definite kernel $\kappa : \mathcal{Z} \times \mathcal{Z} \to \mathbb{R}$ for which the reproducing property $\kappa(z, z') = \langle \phi(z), \phi(z') \rangle_{\mathcal{H}}$ is satisfied (Gretton et al., 2012).

With minibatch size $k$, the maximum variance reduction is achieved by partitioning $\mathcal{D}_s$ into $k$ strata $\mathcal{S} = \{S_h\}_{h=1}^k$ such that $\bigcup S_h = \mathcal{D}_s$, $\bigcap S_h = \varnothing$, and drawing a single instance $(\widetilde{x}_{s,h}, \widetilde{y}_{s,h}) \sim \text{Uniform}(S_h)$ from each (Giles, 2015). Thus, a simple estimate of (2) is

$$\widehat{L}_{\text{MMD}} = \|\widehat{\mu}_{\phi,s} - \widehat{\mu}_{\phi,t}\|_{\mathcal{H}}^2, \tag{3}$$

where $\widehat{\mu}_{\phi,s} = \frac{1}{n_s} \sum_{h=1}^k |S_h| \phi\left(\widetilde{z}_{s,h}\right)$ and $\widetilde{z}_{s,h} = \Theta_F\left(\widetilde{x}_{s,h}\right)$, and similarly for $\widehat{\mu}_{\phi,t}$.

First, we show that to reduce $\text{Var}\left(\widehat{L}_{\text{MMD}}\right)$, a good surrogate objective is to reduce $\text{Var}\left(\widehat{\mu}_{\phi,s}\right) = \mathbb{E}\left[\|\widehat{\mu}_{\phi,s} - \mu_{\phi,s}\|_{\mathcal{H}}^2\right]$ directly. Variance expressions for estimates of the squared MMD have been derived in various prior work (Bounliphone et al., 2016; Liu et al., 2020a; Sutherland & Deka, 2022). However, to keep our analysis straightforward, we use a simpler expression that assumes the empirical mean embeddings are normally distributed. To proceed, let $\Sigma_{\widehat{\mu}_{\phi,s}}$ and $\Sigma_{\widehat{\mu}_{\phi,t}}$ denote the covariance matrices of the respective empirical mean embeddings.

**Lemma 1.** *Assume $\widehat{\mu}_{\phi,s} \sim \mathcal{N}\left(\mu_{\phi,s}, \Sigma_{\widehat{\mu}_{\phi,s}}\right)$ and $\widehat{\mu}_{\phi,t} \sim \mathcal{N}\left(\mu_{\phi,t}, \Sigma_{\widehat{\mu}_{\phi,t}}\right)$ are independent random vectors, and that the relevant conditions for normality under the central limit theorem are satisfied. Then, $\widehat{\mu}_{\phi,s} - \widehat{\mu}_{\phi,t} \sim \mathcal{N}\left(m, \Sigma_{\widehat{\mu}_{\phi,s}} + \Sigma_{\widehat{\mu}_{\phi,t}}\right)$, where $m = \mu_{\phi,s} - \mu_{\phi,t}$, and $\widehat{L}_{\text{MMD}}$ follows a generalised chi-squared distribution with variance (Seber, 2007)*

$$\text{Var}\left(\widehat{L}_{\text{MMD}}\right) = 2\,\text{Tr}\left(\left(\Sigma_{\widehat{\mu}_{\phi,s}} + \Sigma_{\widehat{\mu}_{\phi,t}}\right)^2\right) + 4m^T\left(\Sigma_{\widehat{\mu}_{\phi,s}} + \Sigma_{\widehat{\mu}_{\phi,t}}\right)m. \tag{4}$$

The only quantity dependent on the stratification of $\mathcal{D}_s$ is $\Sigma_{\widehat{\mu}_{\phi,s}}$. Therefore, we consider the variance as a function of $\Sigma_{\widehat{\mu}_{\phi,s}}$ with the remaining variables constant. $\text{Var}\left(\widehat{L}_{\text{MMD}}\right)$ and $\text{Var}\left(\widehat{\mu}_{\phi,s}\right)$ will have the same minimisers if they are related by a monotonic function, which occurs when $\Sigma_{\widehat{\mu}_{\phi,s}}$ is isotropic.

**Theorem 1.** *Assume $\mathcal{H}$ has finite dimensionality $d$ and $\Sigma_{\widehat{\mu}_{\phi,s}} = \sigma^2 I$ for some positive scalar $\sigma^2$. Then,*

$$\arg\min_{\mathcal{S}} \text{Var}\left(\widehat{L}_{\text{MMD}}\right) = \arg\min_{\mathcal{S}} \text{Var}\left(\widehat{\mu}_{\phi,s}\right). \tag{5}$$

*Proof.* See Appendix A. $\qquad\qquad\qquad\qquad\qquad\qquad\qquad\qquad\qquad\qquad\qquad\qquad\qquad\qquad\qquad\square$

For the anisotropic case, the relation is not monotonic and so the special case in Theorem 1 no longer holds. However, by characterising the degree of monotonicity using Spearman's rank correlation $\rho$, we can derive expressions for the expected and worst-case errors of the surrogate solution, in terms of $\rho$, $n_s$, $k$, and the growth rate of $\text{Var}\left(\widehat{L}_{\text{MMD}}\right)$ with respect to the solution rankings.

To proceed, let $f(\mathcal{S}) = \text{Var}\left(\widehat{L}_{\text{MMD}}\right)$ and $g(\mathcal{S}) = \text{Var}\left(\widehat{\mu}_{\phi,s}\right)$ be the target and surrogate partitioning objectives as functions of the dataset partitioning. Define the order statistic $\mathcal{S}_{(i)} = \arg\left\{f(\mathcal{S})_{(i)}\right\}$ for all possible partitionings, e.g., $\mathcal{S}_{(1)} = \arg\min_{\mathcal{S}} f(\mathcal{S})$, $\mathcal{S}_{(2)}$ is the second-best partitioning and so

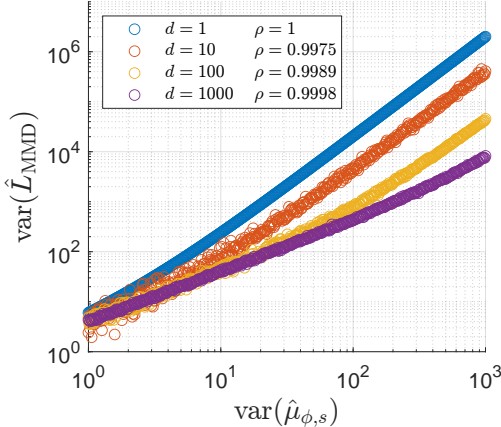

Figure 1: The relationship between the target ($\operatorname{Var}\left(\widehat{L}_{\mathrm{MMD}}\right)$) and surrogate ($\operatorname{Var}\left(\widehat{\mu}_{\phi,s}\right)$) partitioning objectives, for a range of feature dimensionalities $d$.

on. Then define $p = \operatorname{Stirling}(n_s, k)$ the number of such partionings, given by the Stirling partition number. Given a Spearman coefficient $\rho$ between $f$ and $g$, define also a rank displacement $\delta_i = \operatorname{rank}\left(g\left(\mathcal{S}\right)_{(i)}\right) - \operatorname{rank}\left(f\left(\mathcal{S}\right)_{(i)}\right) = \operatorname{rank}\left(g\left(\mathcal{S}\right)_{(i)}\right) - i$, such that, from the formula for Spearman correlation, $\sum \delta_i^2 = \frac{p\left(p^2-1\right)}{6}(1-\rho)$. Finally, let $\mathcal{S}_{(l)} = \arg\min_{\mathcal{S}} g\left(\mathcal{S}\right)$ be the minimiser of $g\left(\mathcal{S}\right)$ which has rank $l$ on $f\left(\mathcal{S}\right)$ – i.e., the best partitioning on $g$ is the $l$th best partitioning on $f$.

**Theorem 2** (Worst-case error bound). *Assume the growth rate of the sorted $f\left(\mathcal{S}\right)_{(i)}$ is bounded by some constant $K$, such that, for any indices $i, j$, $f\left(\mathcal{S}\right)_{(i)} - f\left(\mathcal{S}\right)_{(j)} \leq K(i-j)$. Then, the error incurred by minimising $g$ instead of $f$ satisfies*

$$f\left(\mathcal{S}\right)_{(l)} - f\left(\mathcal{S}\right)_{(1)} \leq K\sqrt{\frac{p\left(p^2-1\right)}{6}(1-\rho)}. \tag{6}$$

*Proof.* As $\delta_1 = l - 1$ and $\delta_1^2 \leq \sum \delta_i^2$, $l$ satisfies $(l-1)^2 \leq \sum \delta_i^2 = \frac{p\left(p^2-1\right)}{6}(1-\rho)$. $\qquad\square$

**Theorem 3** (Expected error). *If the rank displacements are assumed to be homogeneous, then the error is*

$$f\left(\mathcal{S}\right)_{(l)} - f\left(\mathcal{S}\right)_{(1)} = K\sqrt{\frac{p^2-1}{6}(1-\rho)}. \tag{7}$$

*Proof.* With homogeneous displacements, $\delta_1^2 = \sum \delta_i^2 / p$, and so $(l-1)^2 = \frac{p^2-1}{6}(1-\rho)$. $\qquad\square$

Figure 1 shows the relationship between $\operatorname{Var}\left(\widehat{L}_{\mathrm{MMD}}\right)$ and $\operatorname{Var}\left(\widehat{\mu}_{\phi,s}\right)$ for randomly generated covariances $\Sigma_{\widehat{\mu}_{\phi,s}}$, with $m = 1$, $\Sigma_{\widehat{\mu}_{\phi,t}} = 0$, and 4 different values of $d$. Note that for $d = 1$, Theorem 1 holds and so $\rho = 1$ exactly, meaning the errors defined in Theorems 2 and 3 are nil. In the remaining cases, the correlations are all $> 0.99$, validating our choice of surrogate clustering objective.

To solve $\arg\min_{\mathcal{S}} \operatorname{Var}\left(\widehat{\mu}_{\phi,s}\right)$, observe that

$$\operatorname{Var}\left(\widehat{\mu}_{\phi,s}\right) = \operatorname{Var}\left(\frac{1}{n_s}\sum_{h=1}^{k}|S_h|\,\phi\left(\widetilde{z}_{s,h}\right)\right) = \frac{1}{n_s^2}\sum_{i=1}^{k}|S_h|^2\operatorname{Var}\left(\phi\left(\widetilde{z}_{s,h}\right)\right) = \frac{1}{n_s^2}\sum_{h=1}^{k}|S_h|\sum_{z\in S_h}\|\phi(z)-\mu_{\phi,h}\|_{\mathcal{H}}^2, \tag{8}$$

where $\mu_{\phi,h} = \frac{1}{|S_h|} \sum_{z \in S_h} \phi(z)$ is the stratum mean. Thus, we have the optimisation

$$\arg\min_{\mathcal{S}} \sum_{h=1}^{k} |S_h| \sum_{z \in S_h} \|\phi(z) - \mu_{\phi,h}\|_{\mathcal{H}}^2, \tag{9}$$

which can be seen as a kernel k-means-style clustering problem with dynamically weighted clusters. Compared to ordinary (kernel) k-means clustering, this objective imposes a greater penalty on larger clusters, and thus encourages (but does not strictly enforce) the clusters to be of balanced sizes.

### 2.3 Optimal stratification for CORAL

The CORAL loss is defined as

$$L_{\text{CORAL}} = \|R_s - R_t\|_F^2, \tag{10}$$

where $R_s$ and $R_t$ are the covariance matrices of $z_s$ and $z_t$ respectively. An estimate using stratified sampling is

$$\widehat{L}_{\text{CORAL}} = \left\|\widehat{R}_s - \widehat{R}_t\right\|_F^2, \tag{11}$$

where $\widehat{R}_s = \frac{1}{n_s - 1} \sum_{h=1}^{k} |S_h| (\widetilde{z}_{s,h} - \widehat{\mu}_s) (\widetilde{z}_{s,h} - \widehat{\mu}_s)^T$, $\widehat{\mu}_s = \frac{1}{n_s} \sum_{h=1}^{k} |S_h| \widetilde{z}_{s,h}$, and similarly for $\widehat{R}_t$ and $\widehat{\mu}_t$.

As previously, $\widehat{R}_s$ and $\widehat{R}_t$ can be assumed to be Gaussian and so Theorem 1 implies that to minimise $\text{Var}\left(\widehat{L}_{\text{CORAL}}\right)$, a good surrogate is to minimise $\text{Var}\left(\widehat{R}_s\right)$ and $\text{Var}\left(\widehat{R}_t\right)$ directly. However, unlike previously, $\text{Var}\left(\widehat{R}_s\right)$ is not a convenient clustering objective, since $\widehat{R}_s$ cannot be expressed as a kernel mean embedding. Therefore, we propose instead to minimise the variance of

$$\widehat{R}'_s = \frac{1}{n_s} \sum_{h=1}^{k} |S_h| (\widetilde{z}_{s,h} - \mu_s) (\widetilde{z}_{s,h} - \mu_s)^T, \tag{12}$$

which is the sample covariance of $z_s$ in the hypothetical case where $\mu_s$ is known.

Again, we can assess the validity of this surrogate objective by estimating its rank correlation with the $\text{Var}\left(\widehat{R}_s\right)$. Given population parameters $R_s$ and $\mu_s$, $\text{Var}\left(\widehat{R}_s\right)$ and $\text{Var}\left(\widehat{R}'_s\right)$ can be estimated for arbitrary distributions and dimensionality using Monte Carlo simulations. For the special case where $\widetilde{z}_{s,h}, h = 1, \ldots, k$ are Gaussian scalars, an exact relation can also be derived (Theorem 4).

**Theorem 4.** *Assume $\widetilde{z}_{s,h} \sim \mathcal{N}(\mu_h, \Sigma_h), h = 1, \ldots, k$, are independent random scalars, and let $\widehat{R}_s$ and $\widehat{R}'_s$ be sample covariance matrices of $z_s$ as defined. Then,*

$$\text{Var}\left(\widehat{R}_s\right) = \gamma^2 \left(\text{Tr}\left((AC\Sigma)^2\right) + 4\mu^T AC\Sigma AC\mu\right) \tag{13}$$

$$\text{Var}\left(\widehat{R}'_s\right) = 2\,\text{Tr}\left((A\Sigma)^2\right) + 4\mu^T AC\Sigma AC\mu \tag{14}$$

$$\text{Var}\left(\widehat{R}_s\right) = \gamma^2 \left(\text{Var}\left(\widehat{R}'_s\right) + 2\,\text{Tr}\left(A^2\Sigma\right)^2 - 4\,\text{Tr}\left(A^3\Sigma^2\right)\right) \tag{15}$$

*where $\mu = (\mu_1, \ldots, \mu_k)^T$, $\Sigma = \text{diag}((\Sigma_1, \ldots, \Sigma_k))$, $A = \frac{1}{n_s}\text{diag}((|S_1|, \ldots, |S_k|))$, $\gamma = \frac{n_s}{n_s - 1}$ is a bias correction factor, and $C = I - JA$ is the weighted centering matrix, with $J$ the square matrix of ones.*

*Proof.* See Appendix B. □

Figure 2 shows the relationship between $\text{Var}\left(\widehat{R}_s\right)$ against $\text{Var}\left(\widehat{R}'_s\right)$ for a range of dimensionalities of $\mathcal{Z}$ and 2 different distributions, the normal distribution (above) and the lognormal distribution (below), to show an asymmetric non-Gaussian case. We set $k = 8$ and the cluster size ratio $1, \ldots, k$. The rank correlation is near-monotonic in all cases, suggesting that $\text{Var}\left(\widehat{R}'_s\right)$ is indeed a valid surrogate objective.

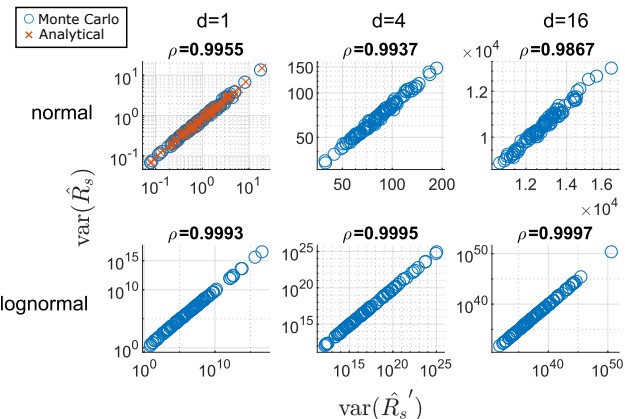

Figure 2: The relationship between the target ($\mathrm{Var}\left(\widehat{R}_s\right)$) and surrogate ($\mathrm{Var}\left(\widehat{R}'_s\right)$) partitioning objectives, for a range of feature dimensionalities $d$, and two distributions.

Note that

$$
\begin{aligned}
\mathrm{Var}\left(\widehat{R}'_s\right) &= \mathrm{Var}\left(\frac{1}{n_s}\sum_{h=1}^{k}|S_h|\left(\widetilde{z}_{s,h}-\mu_s\right)\left(\widetilde{z}_{s,h}-\mu_s\right)^T\right) \\
&= \frac{1}{n_s^2}\sum_{h=1}^{k}|S_h|^2\,\mathrm{Var}\left(\left(\widetilde{z}_{s,h}-\mu_s\right)\left(\widetilde{z}_{s,h}-\mu_s\right)^T\right),
\end{aligned}
\tag{16}
$$

and thus we obtain the same dynamically-weighted kernel k-means problem (9) as before, with the specific mapping $\phi_c(z) = (z-\mu_s)(z-\mu_s)^T$, and an associated kernel $\kappa_c(z,z') = \left(\left(z-\mu_s\right)^T\left(z'-\mu_s\right)\right)^2$.

## 2.4 Optimisation algorithm

Equation (9) can be solved to a local minimum by applying a Lloyd's-style alternating optimisation algorithm (Lloyd, 1982) along with the kernel trick (Algorithm 1).

---
**Algorithm 1** Dynamically-weighted kernel k-means clustering
---
1: **Initialisation:** Use kernel k-means++ (Arthur & Vassilvitskii, 2007) to obtain an initial set of assignments. Then, alternate between Steps 2 and 3 until convergence or max iterations reached.
2: **Distance Update:** Compute the distance matrix $D \in \mathbb{R}^{n_s \times k}$ from each datapoint to the centroid of each cluster using the kernel trick.
3: **Dynamically Weighted Assignment:** Compute the one-hot cluster assignment matrix $U \in \{0,1\}^{n_s \times k}$ that assigns each point to exactly one of the $k$ clusters.

---

$U$ is the solution to the quadratic program

$$
\arg\min_{U}\sum_{i,j}\left[U_{ij}D_{ij}\sum_{i}U_{ij}\right] \quad \text{subject to } \ 0 \le U_{ij} \le 1,\ \sum_{j}U_{ij} = 1.
\tag{17}
$$

Note that we are able to relax the binary constraint $U_{ij} \in \{0,1\}$ as Hoffman's sufficient conditions for total unimodularity (Heller & Tompkins, 1956) are met, meaning the program will always have integer solutions without having to enforce them explicitly. Since the Hessian of (17) is indefinite in general, this problem is nonconvex and thus finding the global minimum is NP-hard. Although the problem as currently defined could be readily input to a gradient-based interior point method (to find a local minimum), these have $O\left((kn_s)^3\right)$ complexity, and we found empirically that this is only practical up to a maximum of a few hundred data

points. Instead, by considering the specific structure of the problem, we propose in this section a scalable heuristic that can find a local minimum in $O(kn_s)$ time. The idea is to construct $U$ incrementally row-by-row using a greedy strategy, weighting the clusters using interim values for the cluster sizes (Algorithm 2).

Greedy algorithms are susceptible to local minima and so a single run of Algorithm 2 is likely to give a poor solution. However, the algorithm can be randomised by permuting the order in which the rows of $U$ are filled. Given the speed (each iteration simply finds the minimum value of a list), a large number of randomised trials can be performed and then the lowest-cost solution selected. We further found that by parallelising row assignments, more random trials are possible in a given time budget, and this improves the best selected solution, even though the average solution is worse (since the $n_j$ cannot be updated between parallel assignments).

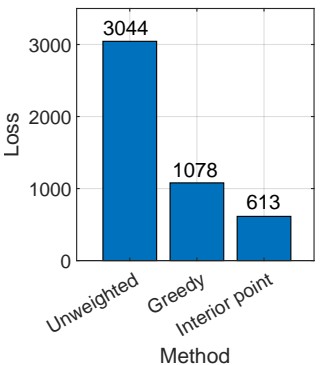

(a) Loss comparison with two alternative approaches.

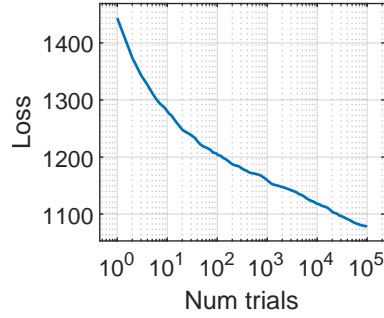

(b) Loss vs number of randomised trials.

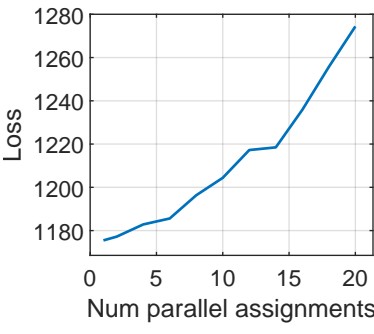

(c) Loss vs number of parallel assignments.

Figure 3: The performance characteristics of Algorithm 2.

The performance characteristics of Algorithm 2 are shown in Figure 3. The input comprises Euclidian distances between randomly sampled embeddings of the Humpbacks dataset (described in the subsequent section). We use a small problem with $n_s = 100$ and $k = 16$ which allows us to compare performance with the interior point solution. Figure 3a compares the solutions found by Algorithm 2 (10 parallel assignments, $10^5$ random trials), a commercial interior point solver (The MathWorks Inc., 2021), as well as an unweighted assignment which simply assigns each point to its closest centroid as in the standard k-means algorithm. Figure 3b shows the effect of varying the number of random trials between 1 and $10^5$, with the number of parallel assignments fixed at 10. Conversely, Figure 3c shows the effect of varying the number of parallel assignments between 1 and 20, with the number of random trials fixed at 100.

An ablation study illustrating the effect of each novel component of our sampling method is shown in Figure 4. The plot shows how $\text{Var}(\widehat{\mu}_{\phi,s})$ varies with $k$ for 4 different sampling strategies: uniform random sampling, stratified sampling with linear k-means clustering, stratified sampling with kernel k-means clustering, and finally Algorithm 1 which dynamically weights the clusters. The simulations use 1,000 feature embeddings from the Humpbacks dataset using a radial basis function kernel with unit bandwidth. It is observed that the gap between the uniform and stratified sampling strategies increases with $k$, with around a 50-fold reduction in variance for $k = 256$. Conversely, the reduction from using kernel k-means rather than linear k-means is only evident for smaller batch sizes, with the two more or less overlapping for larger $k$.

## 2.5 Overall training strategy

Although the optimal stratification depends on the network weights, it is not necessary to reconstruct the strata at every training iteration. This is because, if the network weights at consecutive training iterations are in a locally smooth region of the parameter space, the structure of the embeddings will be preserved between iterations (Liu et al., 2020b). Based on this, Liu et al. (2020b) propose a low-cost criterion to determine when to re-cluster the data. However, for simplicity, for the subsequent experiments we re-cluster

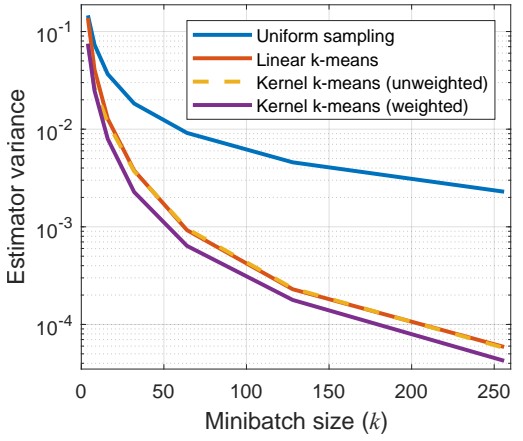

Figure 4: Estimator variance vs minibatch size for Algorithm 1 (purple) vs 3 ablations.

**Algorithm 2** Greedy incremental construction for dynamically weighted assignments

**Require:** $D \in \mathbb{R}^{n_s \times k}$
**Ensure:** $U \in \{0,1\}^{n_s \times k}, \sum_j U_{ij} = 1$
1: $U \leftarrow 0_{n_s \times k}$
2: $n_1, \ldots, n_k \leftarrow 0$ ⊳ Interim cluster sizes
3: **for all** $i \in \{1, \ldots, n_s\}$ **do**
4: $\quad j \leftarrow \arg\min_j D_{ij}(n_j + 1)$
5: $\quad U_{ij} \leftarrow 1$
6: $\quad n_j \leftarrow n_j + 1$
7: **end for**
8: **return** $U$

the data periodically every fixed $T$ iterations, where $T$ is thus a trade-off between the computational burden and the degree of variance reduction achieved.

## 3 Experiments

In this section, the proposed method is evaluated in realistic training conditions to assess whether the reduction in variance observed in Figure 4 translates to an increase in test-domain accuracy. Experiments are conducted using the DomainBed framework (Gulrajani & Lopez-Paz, 2021) on the following 4 domain shift datasets.

**Camelyon17-WILDS** (Bándi et al., 2019; Koh et al., 2021) tumour detection in microscopic tissue images across samples from different hospitals. This benchmark comprises a single training-evaluation split, 5 domains, and 14,000 examples.

**Humpbacks** (Napoli & White, 2023) detection of humpback whale vocalisations in underwater acoustic recordings across data from different acoustic monitoring programs. This benchmark comprises 4 data splits, 4 domains and 8,000 examples.

**Spawrious** (Lynch et al., 2023) classification of 4 dog breeds across images with different background environments (desert, jungle, snow etc.). This benchmark comprises 6 data splits of varying difficulty, 6 domains and 18,664 examples.

**Office-Home** (Venkateswara et al., 2017) image classification of 65 categories of everyday objects with different image styles (Art, Clipart, Product, and Real World). This benchmark comprises 12 data splits, 4 domains and 15,500 examples.

For Spawrious, Camelyon17, and Office-Home, we use a ResNet-18 (He et al., 2015) backbone pre-trained on ImageNet. For Humpbacks, we use the audio frontend and architecture described in Napoli & White (2023). The domain discrepancies are measured between the union of all training data and a held-out subset of the evaluation set. For the MMD, we use a radial basis function (RBF) mixture kernel (Li et al., 2018), given by $\kappa(z, z') = \sum_{\gamma \in \mathcal{G}} e^{-\gamma \|z - z'\|^2}$, with $\mathcal{G} = \{0.001, 0.01, 0.1, 1, 10\}$. With reference to Section 2.4, we run Algorithm 2 with 100 random trials and 10 parallel assignments. Based on empirical observations from Liu et al. (2020b), we choose $T = 100$ for these experiments.

Models are trained using the Adam optimiser (Kingma & Ba, 2014) for 3,000 iterations. Hyperparameters, including the learning rate, weight decay, minibatch size $k$, and trade-off parameter $\lambda$, are tuned with a random search of size 10 using an in-distribution (training domain) validation set, independently for each sampler. In particular, the random search distributions for $k$ and $\lambda$ (which are the hyperparameters with

Table 1: Average test accuracy for each dataset and training algorithm. ERM is included as an ablation but is not part of the main results comparison.

(a) Camelyon17

| Sampler | ERM | CORAL | MMD |
|---|---|---|---|
| Uniform random | $91.6 \pm 0.7$ | $93.1 \pm 0.5$ | $94.5 \pm 0.7$ |
| k-means++ | $92.7 \pm 0.7$ | $94.0 \pm 0.3$ | $94.5 \pm 0.4$ |
| DPP | $90.7 \pm 0.4$ | $94.0 \pm 0.3$ | $94.9 \pm 0.3$ |
| k-means (inputs) | $91.8 \pm 0.9$ | $93.2 \pm 0.4$ | $94.3 \pm 0.3$ |
| k-means (features) | $91.5 \pm 1.3$ | $94.5 \pm 0.2$ | $95.0 \pm 0.4$ |
| VaRDASS | $91.8 \pm 0.6$ | $\mathbf{94.6 \pm 0.3}$ | $\mathbf{95.2 \pm 0.6}$ |

(b) Humpbacks

| Sampler | ERM | CORAL | MMD |
|---|---|---|---|
| Uniform random | $84.2 \pm 1.1$ | $85.3 \pm 1.5$ | $87.7 \pm 1.3$ |
| k-means++ | $84.8 \pm 1.1$ | $89.1 \pm 1.3$ | $88.4 \pm 1.5$ |
| DPP | $83.9 \pm 1.0$ | $\mathbf{89.5 \pm 1.4}$ | $90.3 \pm 1.5$ |
| k-means (inputs) | $82.8 \pm 1.6$ | $85.0 \pm 1.1$ | $89.0 \pm 1.2$ |
| k-means (features) | $84.2 \pm 1.1$ | $87.5 \pm 1.2$ | $89.3 \pm 1.2$ |
| VaRDASS | $83.0 \pm 1.5$ | $88.3 \pm 1.3$ | $\mathbf{92.0 \pm 0.5}$ |

(c) Spawrious

| Sampler | ERM | CORAL | MMD |
|---|---|---|---|
| Uniform random | $58.6 \pm 0.5$ | $62.5 \pm 0.6$ | $64.4 \pm 1.1$ |
| k-means++ | $55.7 \pm 1.0$ | $63.5 \pm 1.2$ | $67.1 \pm 1.7$ |
| DPP | $58.3 \pm 0.6$ | $65.8 \pm 0.9$ | $65.4 \pm 1.3$ |
| k-means (inputs) | $58.8 \pm 0.6$ | $57.7 \pm 1.3$ | $62.6 \pm 0.9$ |
| k-means (features) | $57.9 \pm 0.6$ | $65.2 \pm 0.9$ | $71.4 \pm 1.8$ |
| VaRDASS | $58.6 \pm 0.7$ | $\mathbf{67.7 \pm 0.8}$ | $\mathbf{71.6 \pm 2.2}$ |

(d) Office-Home

| Sampler | ERM | CORAL | MMD |
|---|---|---|---|
| Uniform random | $43.6 \pm 0.3$ | $47.3 \pm 0.3$ | $44.3 \pm 0.3$ |
| k-means++ | $43.4 \pm 0.3$ | $44.8 \pm 0.4$ | $43.1 \pm 0.7$ |
| DPP | $44.6 \pm 0.4$ | $42.1 \pm 0.3$ | $\mathbf{45.4 \pm 0.3}$ |
| k-means (inputs) | $42.2 \pm 0.5$ | $42.1 \pm 0.2$ | $44.6 \pm 0.3$ |
| k-means (features) | $43.8 \pm 0.5$ | $43.4 \pm 0.5$ | $44.4 \pm 0.2$ |
| VaRDASS | $43.6 \pm 0.3$ | $\mathbf{50.7 \pm 0.2}$ | $44.6 \pm 0.4$ |

the largest effect on variance) are $k \sim 2^{\mathrm{Uniform}(3,7)}$ and $\lambda \sim 10^{\mathrm{Uniform}(-1,1)}$. The entire set of experiments is repeated 5 times for reproducibility, using different random seeds for hyperparameters, weight initialisations, and dataset splits. All other hyperparameter choices and training details follow the DomainBed default options – see Gulrajani & Lopez-Paz (2021) for full details. Note that some variance reduction at the gradient level is already achieved for all models through the use of Adam. In addition, shrinkage is also implicitly applied through the tuning of $\lambda$.

In addition to uniform random sampling, five variance-reduced sampling strategies are compared. This includes two diversity-based samplers (Napoli & White, 2024b): the k-means++ algorithm (Arthur & Vassilvitskii, 2007), and the determinantal point process (DPP) (Zhang et al., 2017). Then, three variants of stratified sampling are compared, with different stratification strategies: k-means clustering on the input data (Zhao & Zhang, 2014); k-means clustering on $z_s, z_t$ (an ablation of our method); and VaRDASS, which

Table 2: Average test accuracy of VaRDASS compared to additional UDA methods.

| Method | Camelyon17 | Humpbacks | Spawrious | Office-Home |
|---|---|---|---|---|
| ERM | 91.6 ± 0.7 | 84.2 ± 1.1 | 58.6 ± 0.5 | 43.6 ± 0.3 |
| DANN | 93.2 ± 0.2 | 72.6 ± 3.4 | 67.5 ± 1.3 | 42.0 ± 0.8 |
| CDAN | 91.4 ± 0.5 | 75.4 ± 1.7 | 69.1 ± 1.6 | 42.7 ± 0.5 |
| CDAN + SDAT | 90.9 ± 0.8 | 71.8 ± 2.2 | 69.6 ± 1.8 | 43.5 ± 0.6 |
| CDAN + ELS | 92.1 ± 0.5 | 75.8 ± 1.3 | 70.6 ± 1.9 | 43.1 ± 0.5 |
| ARM | 93.0 ± 1.1 | 84.2 ± 1.4 | 57.0 ± 0.7 | 44.1 ± 0.3 |
| MCC | 94.2 ± 0.6 | 82.7 ± 2.2 | 59.9 ± 1.2 | 45.6 ± 0.2 |
| CORAL | 93.1 ± 0.5 | 85.3 ± 1.5 | 62.5 ± 0.6 | 47.3 ± 0.3 |
| CORAL + VaRDASS | 94.6 ± 0.3 | 88.3 ± 1.3 | 67.7 ± 0.8 | **50.7 ± 0.2** |
| MMD | 94.5 ± 0.7 | 87.7 ± 1.3 | 64.4 ± 1.1 | 44.3 ± 0.3 |
| MMD + VaRDASS | **95.2 ± 0.6** | **92.0 ± 0.5** | **71.6 ± 2.2** | 44.6 ± 0.4 |

Table 3: Average wall-clock training times by dataset (seconds).

| Sampler | Camelyon17 | Humpbacks | Spawrious | Office-Home |
|---|---|---|---|---|
| Uniform random | 180 ± 5 | 30 ± 1 | 393 ± 9 | 625 ± 30 |
| k-means++ | 1602 ± 110 | 345 ± 15 | 2312 ± 151 | 1977 ± 65 |
| DPP | 2654 ± 118 | 731 ± 19 | 4123 ± 109 | 2529 ± 120 |
| k-means (features) | 670 ± 13 | 145 ± 1 | 1071 ± 6 | 1375 ± 68 |
| VaRDASS | 2136 ± 25 | 528 ± 9 | 2958 ± 21 | 2163 ± 38 |

clusters $z_s, z_t$ using Algorithm 1. Table 1 shows the average test accuracy and standard errors over the data splits and repeats, for each dataset. Breakdowns by data split for Humpbacks, Spawrious, and Office-Home are given in Appendix C, Tables 5, 6, and 7.

It can be seen that clustering the input data (Zhao & Zhang, 2014) is not effective in improving test domain accuracy, since the required smoothness conditions do not hold for deeper models. Performing simple k-means clustering on the features provides a decent improvement (corroborated by Figure 4, which also shows a good reduction in estimator variance). In addition, both diverse samplers also improve accuracy, but this is inconsistent at times. There does not appear to be a significant difference between k-means++ and the DPP, although the former is computationally faster. However, VaRDASS performs best overall, with the highest accuracy in six out of the eight cases.

In addition to the targeted UDA algorithms, the tables also report test-domain accuracy in the case of non-adaptive training by empirical risk minimisation (ERM). The performance of ERM does not significantly change by varying the sampler, which indicates that the gains observed in CORAL and MMD are indeed due to the sampler's effect on the adaptation component of the loss.

Table 2 provides a comparison of VaRDASS with the following baseline UDA methods (using uniform random samplers): Domain-Adversarial Neural Network (DANN) (Ganin et al., 2015), Conditional Domain-Adversarial Network (CDAN) (Long et al., 2017), CDAN + Smooth Domain Adversarial Training (SDAT) (Rangwani et al., 2022), CDAN + Environment Label Smoothing (ELS) (Zhang et al., 2023), Adaptive Risk Minimisation (ARM) (Zhang et al., 2020), and Minimum Class Confusion (MCC) (Jin et al., 2020), alongside ERM, CORAL and MMD. It can be seen that VaRDASS performs the best on all four datasets.

Overall wall-clock training times are reported for each sampler and dataset in Table 3. These are averaged over all data splits, hyperparameters, and repeats from the previous experiment, including both CORAL and MMD. It can be seen that our implementation of VaRDASS is around an order of magnitude slower than simple random sampling, and 2-3 times slower than vanilla k-means clustering.

We also investigate whether the discrepancy between the training and test domain features is lower when the variance reduced samplers are used (Appendix D). The domain discrepancy is measured between held-out datapoints which were not seen during training. These results do not demonstrate a difference in the post-

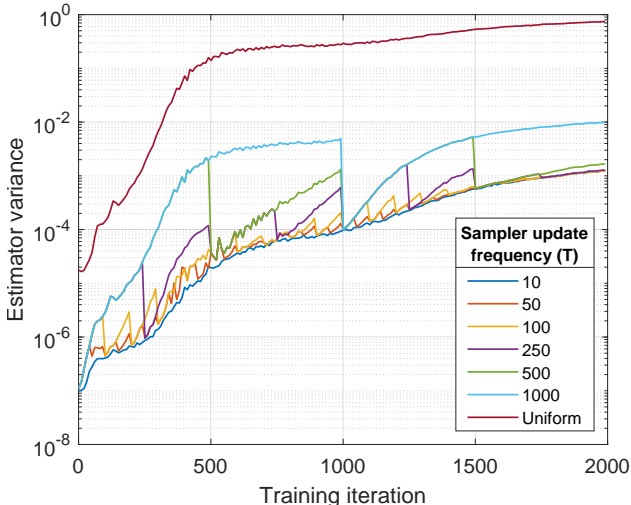

Figure 5: Estimator variance ($\text{Var}\,(\widehat{\mu}_{\phi,s})$) throughout training with VaRDASS for different values of $T$, also compared to uniform random sampling.

Table 4: Average test accuracy of VaRDASS on Humpbacks for different values of $T$.

| Method | 10 | 50 | 100 | 500 | 1000 |
|---|---|---|---|---|---|
| CORAL | $90.3 \pm 2.6$ | $88.2 \pm 3.0$ | $88.3 \pm 1.3$ | $88.3 \pm 2.3$ | $89.8 \pm 2.3$ |
| MMD | $92.2 \pm 1.9$ | $91.7 \pm 1.3$ | $92.0 \pm 0.5$ | $92.3 \pm 1.7$ | $92.4 \pm 1.8$ |

training discrepancy between the standard and variance-reduced samplers. Moreover, all the discrepancy values are already extremely small (two to three orders of magnitude below those obtained under ERM). Thus, the finding is that VaRDASS improves the final target-domain accuracy even though the discrepancy values ultimately converge to similar levels across samplers. This indicates that the primary effect of VaRDASS is to stabilise the optimisation trajectory during training, rather than to obtain a deeper minimum.

### 3.1 Effect of $T$

Figure 5 illustrates how $\text{Var}\,(\widehat{\mu}_{\phi,s})$ varies through one training run of the Humpbacks dataset (using $k = 32$ and a learning rate of $10^{-4}$). It compares the estimator variance achieved by VaRDASS for different values of $T$, as well with a uniform random minibatch sampler. Clearly, since the features distort more with each model update, this means that the less frequently the cluster allocations are updated, the worse these become over time with respect to the clustering objective (9), and so the estimator variance increases. This effect is strongest towards the start of training, since the gradient updates, and thus the feature distortions, are larger. Thus, this implies that adopting an increasing schedule for $T$ could help to reduce unnecessary computation, with negligible detriment to the degree of variance reduction achieved.

On the other hand, the final model test accuracy appears to remain fairly stable across different values of $T$ (Table 4). This result indicates that, in terms of final test accuracy, VaRDASS is rather robust to the choice of $T$, and that even a very low update frequency is enough to boost the model performance.

## 4 Limitations

Like all kernel methods, Algorithm 1 requires constructing a kernel matrix, which has quadratic cost in the size of the training set. Although low-rank approximation methods exist which can improve the scalability to linear (Chitta et al., 2014), this was not investigated in this paper, and the method will likely still be impractical for very large-scale datasets.

k-means clustering is known to be NP-hard and heuristics such as Lloyd's (on which Algorithm 1 is based) only search for local solutions. Thus, even though our derived objective is a good surrogate for minimising variance, there is no guarantee that an improved solution can actually be found. Similarly, the runtime of Lloyd's is known to be superpolynomial in the worst case (Arthur & Vassilvitskii, 2006), although this algorithm remains popular due to its simplicity and the fact it performs well in practice (Arthur et al., 2009; Cordeiro de Amorim & Makarenkov, 2023).

Another problem which can emerge with kernel k-means clustering, especially when $\mathcal{H}$ is high dimensional, is large imbalance in the strata sizes or even singleton strata. This would slow down the convergence rate of the training, but can be dealt with by introducing minimum cluster size constraints to the cluster assignment step.

## 5 Conclusion

This paper introduced VaRDASS, a novel stochastic variance reduction technique for UDA based on stratified sampling, which specifically targets the MMD and CORAL losses. A novel clustering algorithm was proposed to solve this problem and thoroughly analysed. We showed that the method significantly reduces variance in the targeted losses, and that this results in consistent improvements in target domain accuracy. For future work, the method could adopt a strata reconstruction condition as in Liu et al. (2020b), be combined with complementary variance reduction techniques, or be extended to additional UDA or domain generalisation algorithms. Alternative clustering algorithms with superior performance to Lloyd's could also be investigated.

### 5.1 Follow-up work

For the interested reader, a follow-up method which achieves greater variance reduction by permuting the data sampling order itself can be found in Napoli & White (2026).

## 6 Acknowledgements

This work was supported by grants from BAE Systems and the Engineering and Physical Sciences Research Council. The authors acknowledge the use of the IRIDIS High Performance Computing Facility, and associated support services at the University of Southampton, in the completion of this work.

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

## A Proof of Theorem 1

**Theorem 1.** *Assume $\mathcal{H}$ has finite dimensionality $d$ and $\Sigma_{\widehat{\mu}_{\phi,s}} = \sigma^2 I$ for some positive scalar $\sigma^2$. Then,*

$$\arg\min_{\mathcal{S}} \operatorname{Var}\left(\widehat{L}_{\mathrm{MMD}}\right) = \arg\min_{\mathcal{S}} \operatorname{Var}\left(\widehat{\mu}_{\phi,s}\right). \tag{18}$$

*Proof.* Noting that $\operatorname{Var}\left(\widehat{\mu}_{\phi,s}\right) = \sigma^2 d$, we have

$$
\begin{aligned}
\operatorname{Var}\left(\widehat{L}_{\mathrm{MMD}}\right) &= 2\operatorname{Tr}\left(\left(\sigma^2 I + \Sigma_{\widehat{\mu}_{\phi,t}}\right)^2\right) + 4m^T\left(\sigma^2 I + \Sigma_{\widehat{\mu}_{\phi,t}}\right)m \\
&= 2\operatorname{Tr}\left(\sigma^4 I + 2\sigma^2\Sigma_{\widehat{\mu}_{\phi,t}} + \Sigma_{\widehat{\mu}_{\phi,t}}^2\right) + 4\sigma^2 m^T m + 4m^T\Sigma_{\widehat{\mu}_{\phi,t}}m \\
&= 2\sigma^4 d + 4\sigma^2\left(m^T m + \operatorname{Tr}\left(\Sigma_{\widehat{\mu}_{\phi,t}}\right)\right) + 2\operatorname{Tr}\left(\Sigma_{\widehat{\mu}_{\phi,t}}^2\right) + 4m^T\Sigma_{\widehat{\mu}_{\phi,t}}m \\
&= \frac{2}{d}\operatorname{Var}\left(\widehat{\mu}_{\phi,s}\right)^2 + \frac{4}{d}\operatorname{Var}\left(\widehat{\mu}_{\phi,s}\right)\left(m^T m + \operatorname{Tr}\left(\Sigma_{\widehat{\mu}_{\phi,t}}\right)\right) \\
&\quad + 2\operatorname{Tr}\left(\Sigma_{\widehat{\mu}_{\phi,t}}^2\right) + 4m^T\Sigma_{\widehat{\mu}_{\phi,t}}m,
\end{aligned}
\tag{19}
$$

which is monotonically increasing in $\operatorname{Var}\left(\widehat{\mu}_{\phi,s}\right)$ for all fixed $m, \Sigma_{\widehat{\mu}_{\phi,t}}, d$. $\square$

## B Proof of Theorem 4

**Theorem 4.** *Assume $\widetilde{z}_{s,h} \sim \mathcal{N}\left(\mu_h, \Sigma_h\right), h = 1, \ldots, k$ are independent random scalars, and let $\widehat{R}_s$ and $\widehat{R}'_s$ be sample covariance matrices of $z_s$ as defined. Then,*

$$\operatorname{Var}\left(\widehat{R}_s\right) = \gamma^2\left(\operatorname{Tr}\left((AC\Sigma)^2\right) + 4\mu^T AC\Sigma AC\mu\right) \tag{20}$$

$$\operatorname{Var}\left(\widehat{R}'_s\right) = 2\operatorname{Tr}\left((A\Sigma)^2\right) + 4\mu^T AC\Sigma AC\mu \tag{21}$$

$$\operatorname{Var}\left(\widehat{R}_s\right) = \gamma^2\left(\operatorname{Var}\left(\widehat{R}'_s\right) + 2\operatorname{Tr}\left(A^2\Sigma\right)^2 - 4\operatorname{Tr}\left(A^3\Sigma^2\right)\right) \tag{22}$$

*where $\mu = (\mu_1, \ldots, \mu_k)^T$, $\Sigma = \operatorname{diag}\left((\Sigma_1, \ldots, \Sigma_k)\right)$, $A = \frac{1}{n_s}\operatorname{diag}\left((|S_1|, \ldots, |S_k|)\right)$, $\gamma = \frac{n_s}{n_s - 1}$ is a bias correction factor, and $C = I - JA$ is the weighted centering matrix, with $J$ the square matrix of ones.*

*Proof.* For scalar $\widetilde{z}_{s,h}$, the covariances can be written as quadratic forms

$$\widehat{R}_s = (CZ)^T A(CZ) = \gamma Z^T ACZ \tag{23}$$

$$\widehat{R}'_s = (Z - \mu_s)^T A\left(Z - \mu_s\right) \tag{24}$$

with $Z = (\widetilde{z}_{s,1}, \ldots, \widetilde{z}_{s,k})^T \sim \mathcal{N}(\mu, \Sigma)$. For $\operatorname{Var}\left(\widehat{R}_s\right)$, the expression is immediate from Seber (2007). For $\operatorname{Var}\left(\widehat{R}'_s\right)$, one has

$$\operatorname{Var}\left(\widehat{R}'_s\right) = 2\operatorname{Tr}\left((A\Sigma)^2\right) + 4\left(\mu - \mu_s\right)^T A\Sigma A\left(\mu - \mu_s\right) \tag{25}$$

and note that $C\mu = \mu - \mu_s$ because $\mu_s = \frac{1}{n_s}\sum_{h=1}^k |S_h|\mu_h$. The result follows.

Expanding $\operatorname{Tr}\left((AC\Sigma)^2\right) = \operatorname{Tr}\left((A(I - JA)\Sigma)^2\right)$ and substituting then gives

$$
\begin{aligned}
\frac{1}{\gamma^2}\operatorname{Var}\left(\widehat{R}_s\right) &= 2\left(\operatorname{Tr}\left((A\Sigma)^2\right) + \operatorname{Tr}\left(A^2\Sigma\right)^2 - 2\operatorname{Tr}\left(A^3\Sigma^2\right)\right) + 4\mu^T AC\Sigma AC\mu \\
&= \operatorname{Var}\left(\widehat{R}'_s\right) + 2\operatorname{Tr}\left(A^2\Sigma\right)^2 - 4\operatorname{Tr}\left(A^3\Sigma^2\right). \qquad \square
\end{aligned}
\tag{26}
$$

## C  Performance breakdown by data split

Table 5: Average test accuracy for Humpbacks by data split.

| Method | Domain 1 | Domain 2 | Domain 3 | Domain 4 | Average |
|---|---|---|---|---|---|
| ERM | 70.3 ± 2.7 | 92.0 ± 1.9 | 78.1 ± 3.0 | 96.2 ± 0.6 | 84.2 ± 1.1 |
| DANN | 60.5 ± 4.2 | 90.1 ± 2.1 | 63.9 ± 6.1 | 76.1 ± 11.1 | 72.6 ± 3.4 |
| CDAN | 61.6 ± 4.2 | 82.2 ± 4.7 | 73.5 ± 3.0 | 84.4 ± 0.6 | 75.4 ± 1.7 |
| CDAN + SDAT | 63.7 ± 3.0 | 81.2 ± 6.7 | 63.6 ± 3.7 | 78.9 ± 3.2 | 71.8 ± 2.2 |
| CDAN + ELS | 62.7 ± 2.4 | 85.6 ± 3.4 | 70.8 ± 2.2 | 83.9 ± 2.3 | 75.8 ± 1.3 |
| ARM | 78.8 ± 3.4 | 95.9 ± 1.0 | 72.4 ± 3.6 | 89.6 ± 2.0 | 84.2 ± 1.4 |
| MCC | 70.0 ± 5.9 | 87.7 ± 4.9 | 76.1 ± 4.1 | 97.1 ± 0.5 | 82.7 ± 2.2 |
| CORAL | 77.2 ± 1.4 | 87.7 ± 3.4 | 83.6 ± 4.2 | 92.7 ± 1.9 | 85.3 ± 1.5 |
| + k-means++ | 79.1 ± 1.7 | **97.8 ± 0.6** | 85.4 ± 4.6 | 93.8 ± 1.0 | 89.1 ± 1.3 |
| + DPP | 81.0 ± 1.9 | 97.4 ± 1.0 | 84.4 ± 5.1 | **94.9 ± 1.2** | **89.5 ± 1.4** |
| + k-means (inputs) | 72.0 ± 2.4 | 93.2 ± 1.5 | 83.0 ± 3.4 | 91.6 ± 0.9 | 85.0 ± 1.1 |
| + k-means (features) | **81.4 ± 0.6** | 89.4 ± 2.3 | **86.1 ± 4.0** | 93.1 ± 1.2 | 87.5 ± 1.2 |
| + VaRDASS | 78.4 ± 2.8 | 95.1 ± 1.6 | 85.3 ± 4.2 | 94.4 ± 1.2 | 88.3 ± 1.3 |
| MMD | 78.3 ± 2.5 | 95.0 ± 1.2 | 83.3 ± 4.2 | 94.3 ± 1.1 | 87.7 ± 1.3 |
| + k-means++ | 79.7 ± 1.9 | 97.6 ± 0.4 | 79.5 ± 5.7 | **96.7 ± 0.3** | 88.4 ± 1.5 |
| + DPP | 83.4 ± 1.0 | 97.9 ± 0.6 | 83.4 ± 5.9 | 96.6 ± 0.6 | 90.3 ± 1.5 |
| + k-means (inputs) | 76.6 ± 2.5 | 95.9 ± 1.4 | 88.1 ± 3.7 | 95.5 ± 0.9 | 89.0 ± 1.2 |
| + k-means (features) | **80.4 ± 1.9** | 97.1 ± 0.6 | 85.7 ± 4.2 | 93.9 ± 0.6 | 89.3 ± 1.2 |
| + VaRDASS | 79.2 ± 1.1 | **98.4 ± 0.5** | **95.5 ± 1.2** | 95.0 ± 1.0 | **92.0 ± 0.5** |

Table 6: Average test accuracy for Spawrious by data split.

| Method | O2O-Easy | O2O-Medium | O2O-Hard | M2M-Easy | M2M-Medium | M2M-Hard | Average |
|---|---|---|---|---|---|---|---|
| ERM | 68.6 ± 1.7 | 62.6 ± 0.8 | 62.1 ± 0.7 | 70.2 ± 1.8 | 45.0 ± 1.3 | 43.0 ± 1.2 | 58.6 ± 0.5 |
| DANN | 91.4 ± 3.0 | 57.1 ± 3.5 | 71.1 ± 3.2 | 91.1 ± 0.1 | 54.8 ± 4.4 | 39.8 ± 3.4 | 67.5 ± 1.3 |
| CDAN | 91.9 ± 1.7 | 57.0 ± 3.0 | 70.3 ± 2.2 | 92.9 ± 0.9 | 58.3 ± 3.8 | 44.3 ± 7.1 | 69.1 ± 1.6 |
| CDAN + SDAT | 92.9 ± 1.4 | 54.3 ± 3.5 | 73.7 ± 6.6 | 83.3 ± 3.0 | 60.3 ± 4.3 | 53.0 ± 6.0 | 69.6 ± 1.8 |
| CDAN + ELS | 89.8 ± 1.9 | 58.4 ± 2.1 | 67.3 ± 1.5 | 89.9 ± 2.2 | 62.3 ± 2.3 | 56.1 ± 10.2 | 70.6 ± 1.9 |
| ARM | 70.2 ± 2.9 | 58.6 ± 2.1 | 60.6 ± 0.2 | 68.7 ± 1.6 | 42.2 ± 1.8 | 41.7 ± 1.0 | 57.0 ± 0.7 |
| MCC | 87.6 ± 1.9 | 51.0 ± 0.8 | 54.0 ± 6.3 | 77.5 ± 2.4 | 46.4 ± 0.5 | 42.7 ± 1.2 | 59.9 ± 1.2 |
| CORAL | 70.7 ± 2.3 | 58.4 ± 1.9 | 64.1 ± 0.6 | 78.6 ± 1.5 | 54.1 ± 1.2 | 49.2 ± 0.7 | 62.5 ± 0.6 |
| + k-means++ | 82.8 ± 3.5 | 58.2 ± 2.4 | 61.4 ± 4.1 | 75.5 ± 2.7 | 54.7 ± 2.7 | 48.6 ± 1.1 | 63.5 ± 1.2 |
| + DPP | 79.8 ± 3.4 | 59.8 ± 2.3 | 67.8 ± 2.0 | 79.6 ± 2.3 | 58.5 ± 1.4 | **49.4 ± 1.9** | 65.8 ± 0.9 |
| + k-means (inputs) | 80.4 ± 5.0 | 59.6 ± 1.5 | 50.8 ± 1.5 | 79.5 ± 0.9 | 49.5 ± 2.1 | 26.3 ± 4.8 | 57.7 ± 1.3 |
| + k-means (features) | 85.8 ± 2.2 | 58.7 ± 1.9 | 60.4 ± 2.4 | **81.8 ± 1.9** | **55.2 ± 2.3** | 49.2 ± 2.2 | 65.2 ± 0.9 |
| + VaRDASS | **90.1 ± 2.2** | **62.2 ± 1.0** | **79.6 ± 2.8** | 77.0 ± 2.7 | 51.7 ± 1.6 | 45.8 ± 0.4 | **67.7 ± 0.8** |
| MMD | 79.2 ± 3.3 | 61.9 ± 1.2 | 65.5 ± 3.4 | 76.2 ± 3.4 | 55.3 ± 3.4 | 48.1 ± 0.7 | 64.4 ± 1.1 |
| + k-means++ | 83.7 ± 6.3 | 58.6 ± 2.4 | 68.4 ± 4.5 | 79.3 ± 2.7 | 60.0 ± 3.0 | 52.5 ± 4.5 | 67.1 ± 1.7 |
| + DPP | 83.6 ± 4.5 | **62.9 ± 0.9** | 63.5 ± 3.2 | 79.1 ± 3.1 | 57.4 ± 4.0 | 45.6 ± 1.7 | 65.4 ± 1.3 |
| + k-means (inputs) | 85.9 ± 2.9 | 58.2 ± 2.2 | 58.8 ± 2.0 | 76.6 ± 2.3 | 50.8 ± 1.5 | 45.5 ± 1.4 | 62.6 ± 0.9 |
| + k-means (features) | 82.1 ± 5.0 | 60.2 ± 2.2 | **78.4 ± 3.6** | 80.0 ± 3.3 | 66.8 ± 4.3 | **60.9 ± 6.3** | 71.4 ± 1.8 |
| + VaRDASS | **94.2 ± 1.8** | 61.5 ± 1.4 | 72.7 ± 4.5 | 76.9 ± 4.3 | **75.9 ± 10.6** | 48.1 ± 5.1 | **71.6 ± 2.2** |

Table 7: Average test accuracy for Office-Home by data split.

| Method | C-A | A-C | P-A | A-P | R-A | A-R | P-C | C-P | R-C | C-R | R-P | P-R | Average |
|---|---|---|---|---|---|---|---|---|---|---|---|---|---|
| ERM | 32.3 | 30.8 | 29.3 | 40.1 | 47.4 | 53.8 | 32.3 | 45.9 | 35.5 | 49.1 | 66.9 | 59.7 | 43.6 ± 0.2 |
| DANN | 30.3 | 33.0 | 27.3 | 38.6 | 47.0 | 54.5 | 32.9 | 42.9 | 38.4 | 42.4 | 62.5 | 53.8 | 42.0 ± 0.8 |
| CDAN | 33.0 | 32.8 | 29.0 | 40.9 | 47.2 | 53.8 | 28.8 | 43.0 | 40.8 | 47.6 | 63.2 | 52.2 | 42.7 ± 0.5 |
| CDAN + SDAT | 30.3 | 32.7 | 26.6 | 40.2 | 50.8 | 55.9 | 31.5 | 43.6 | 39.1 | 46.8 | 67.0 | 56.9 | 43.5 ± 0.6 |
| CDAN + ELS | 28.2 | 33.2 | 28.4 | 41.0 | 48.7 | 55.3 | 30.7 | 45.5 | 40.7 | 48.3 | 64.1 | 52.6 | 43.1 ± 0.5 |
| ARM | 32.9 | 28.4 | 31.6 | 41.5 | 46.2 | 57.3 | 30.1 | 48.6 | 37.7 | 49.9 | 67.8 | 57.7 | 44.1 ± 0.3 |
| MCC | 30.4 | 33.0 | 31.4 | 45.8 | 46.3 | 58.4 | 35.5 | 51.1 | 41.1 | 50.7 | 66.2 | 57.9 | 45.6 ± 0.2 |
| CORAL | 38.6 | 33.1 | 34.8 | 37.9 | 55.9 | 53.6 | 39.6 | 47.7 | 47.3 | 50.7 | 69.8 | 59.1 | 47.3 ± 0.3 |
| + k-means++ | 35.6 | 31.1 | 29.5 | 35.9 | 55.7 | 53.0 | 35.4 | 45.7 | 41.9 | 46.6 | 67.1 | 56.0 | 44.8 ± 0.4 |
| + DPP | 34.2 | 27.5 | 30.9 | 33.5 | 45.9 | 52.4 | 26.6 | 42.3 | 38.0 | 41.6 | 66.8 | 56.0 | 42.1 ± 0.3 |
| + k-means (inputs) | 32.0 | 23.7 | 33.7 | 33.0 | 54.9 | 54.0 | 27.5 | 44.5 | 31.7 | 44.4 | 66.2 | 59.0 | 42.1 ± 0.2 |
| + k-means (features) | 25.8 | 30.5 | 30.2 | 43.1 | 55.9 | 56.5 | 32.7 | 45.9 | 37.7 | 39.6 | 69.0 | 53.4 | 43.4 ± 0.5 |
| + VaRDASS | **40.1** | **35.5** | **35.2** | **43.8** | **57.4** | **57.6** | **43.2** | **52.5** | **49.7** | **55.8** | **73.6** | **63.8** | **50.7 ± 0.2** |
| MMD | 32.9 | 33.6 | 29.1 | 42.9 | 48.0 | 54.1 | 32.1 | 47.7 | 37.3 | 51.0 | 64.6 | 58.3 | 44.3 ± 0.3 |
| + k-means++ | 32.6 | 31.5 | 27.8 | 43.8 | 45.6 | 54.3 | 34.5 | 45.8 | 37.2 | 49.5 | 62.9 | 57.0 | 43.1 ± 0.7 |
| + DPP | **33.1** | 35.4 | **32.4** | 43.5 | **50.0** | **58.2** | 31.8 | **48.0** | 37.3 | 50.1 | 66.7 | **58.5** | **45.4 ± 0.3** |
| + k-means (inputs) | 32.9 | 34.9 | 30.7 | 43.6 | 46.6 | 45.4 | 31.7 | 48.1 | 37.8 | **51.4** | 66.6 | 54.4 | 44.6 ± 0.3 |
| + k-means (features) | 32.8 | **35.6** | 30.7 | 45.5 | 46.0 | 55.6 | 32.9 | 45.8 | 36.7 | 47.8 | 64.8 | 58.3 | 44.4 ± 0.2 |
| + VaRDASS | 31.6 | 33.4 | 31.3 | **45.9** | 48.5 | 53.0 | 30.8 | 47.3 | **38.1** | 49.8 | **66.8** | 58.3 | 44.6 ± 0.4 |

# D   Domain discrepancy post-training

Table 8: Average discrepancy between the training and test domains for each dataset.

(a) Camelyon17

| Sampler | CORAL | MMD |
|---|---|---|
| Uniform random | 0.151 ± 0.074 | **0.135 ± 0.021** |
| k-means (inputs) | **0.093 ± 0.011** | 0.183 ± 0.030 |
| k-means (features) | 0.117 ± 0.084 | 0.260 ± 0.038 |
| VaRDASS | 0.114 ± 0.023 | 0.217 ± 0.042 |

(b) Humpbacks

| Sampler | CORAL | MMD |
|---|---|---|
| Uniform random | 5.712 ± 1.203 | 6.489 ± 0.289 |
| k-means (inputs) | **4.551 ± 1.049** | 6.936 ± 0.380 |
| k-means (features) | 8.984 ± 1.885 | 6.396 ± 0.326 |
| VaRDASS | 8.695 ± 1.533 | **6.008 ± 0.321** |

(c) Spawrious

| Sampler | CORAL | MMD |
|---|---|---|
| Uniform random | 0.281 ± 0.049 | **0.206 ± 0.038** |
| k-means (inputs) | **0.201 ± 0.043** | 0.229 ± 0.041 |
| k-means (features) | 0.270 ± 0.061 | 0.255 ± 0.041 |
| VaRDASS | 0.220 ± 0.034 | 0.317 ± 0.051 |

