# OpenReview forum: "Variance Matters: Improving Domain Adaptation via Stratified Sampling"
_TMLR — Accepted by TMLR_

### Review · Reviewer_iS8e · 2026-02-09

**Summary Of Contributions:**

This paper addresses the challenge of high variance in domain discrepancy estimates during Unsupervised Domain Adaptation. The authors propose VaRDASS, a specialized stochastic variance reduction technique for UDA.
### Key contributions include:
- **Theoretical Derivations**: The authors derive specific stratification objectives for two widely used discrepancy measures
- **Optimality Proofs**: For MMD, the authors prove that their proposed objective is theoretically optimal for minimizing variance under certain assumptions.
- **Algorithmic Innovation**: A practical, k-means style optimization algorithm is introduced to solve the stratification problem in $O(kn_s)$ time, making it more scalable than standard interior-point methods.
- **Empirical Validation**: Experiments on three domain shift datasets (Camelyon17, Humpbacks, and Spawrious) demonstrate that VaRDASS reduces discrepancy estimation error and improves target domain performance over uniform sampling and other baseline samplers.

### Strengths:
- The paper tackles a well-documented problem (noisy discrepancy estimates) with a novel approach.
- Solid theoretical grounding, including error bounds and Spearman's rank correlation analysis to justify surrogate objectives.
- Comparisons to various UDA baselines and diverse sampling methods.

### Weaknesses:
- **Outdated Baselines for a Well-Documented Problem**: While the paper addresses a well-recognized challenge in UDA (noisy discrepancy estimates), the comparative evaluation leans heavily on older baselines (e.g., DANN from 2015, CDAN from 2017). In a well-documented field, it is critical to demonstrate that the proposed sampling strategy remains competitive or complementary when applied to more recent UDA architectures.
- Computational overhead associated with constructing kernel matrices for large datasets.
- Sensitivity to the re-clustering frequency parameter $T$.

**Audience:**

Yes

**Audience Explanation:**

Domain adaptation is a critical area of research for the TMLR audience, particularly those focused on the robustness and deployment of machine learning models. The finding that "variance matters" in UDA and that it can be mitigated through smarter sampling, rather than just architectural changes. This is a valuable insight for both theorists and practitioners.

**Broader Impact Concerns:**

N/A. The work focuses on improving the efficiency and robustness of standard machine learning algorithms. It uses established public benchmarks (Camelyon17, Humpbacks, Spawrious) and does not introduce new ethical risks or data privacy concerns.

**Claims And Evidence:**

Yes

**Claims Explanation:**

The authors provide a balanced mix of theoretical and empirical evidence to support their claims, including:
- **Variance Reduction**: Figure 4 provides clear evidence that VaRDASS achieves a significant reduction in estimator variance compared to uniform sampling.
- **Accuracy Improvements**: Table 1 and Table 2 show that VaRDASS consistently yields the highest test accuracy across most datasets and discrepancy measures, outperforming established methods like DANN, CDAN, and ARM.
- **Surrogate Validity**: Figure 1 and Figure 2 empirically validate that the simplified surrogate objectives are highly correlated with the true variance objectives ($\rho > 0.98$), justifying their use in the practical algorithm.

**Requested Changes:**

- **Adding Recent Baselines for Comparison**. The submission addresses the problem of high-variance gradients in UDA, which is a well-documented challenge. However, the current evaluation primarily compares VaRDASS against older architectures such as DANN (2015) and CDAN (2017). To provide convincing evidence that this sampling strategy offers a distinct advantage in the current research landscape, the authors must: 1) Include at least two modern UDA baselines from the 2022–2025 period. 2) Demonstrate that the variance reduction benefits of VaRDASS are complementary to, or outperform, the implicit regularization found in these newer methods.
- **Computational Cost Analysis**: While the authors propose a greedy heuristic with $O(kn_s)$ complexity , the actual training time overhead (including kernel matrix construction and periodic re-clustering) remains unclear. Provide a comparative table of wall-clock training times across different datasets to justify the efficiency-performance trade-off.
- **Hyperparameter Sensitivity ($T$)**: Provide a breakdown of how the sampler update frequency $T$ affects the final target domain accuracy, rather than just the estimator variance.
- **Post-Training Discrepancy Discussion**: Elaborate on why the post-training discrepancy results were inconclusive. Specifically, clarify if this suggests that VaRDASS functions primarily as a stabilizer for the optimization trajectory rather than as a tool to reach a deeper global minimum.

---

> ### Author Response · Authors · 2026-03-06
>
> Thank you for your feedback.
>
> 1. We have added results on two modern baselines: Smooth Domain Adversarial Training [1] and Environment Label Smoothing [2]. Both of these are based on advanced regularisation methods, as you have mentioned.
> 2. We have added a comparison of wall-clock training times (Table 3).
> 3. We have added an analysis of the effect of $T$ on the final target domain accuracy (Table 4).
> 4. Thank you for your insight regarding the post-training discrepancy results, which we believe is a valuable and valid interpretation. We have expanded the discussion to include this explanation.
>
> [1] Harsh Rangwani, Sumukh K Aithal, Mayank Mishra, Arihant Jain, and R Venkatesh Babu. A Closer Look at Smoothness in Domain Adversarial Training. ICML, 2022.
>
> [2] YiFan Zhang, Xue Wang, Jian Liang, Zhang Zhang, Liang Wang, Rong Jin, and Tieniu Tan. Free Lunch for Domain Adversarial Training: Environment Label Smoothing. ICLR, 2023.

---

### Review · Reviewer_JMmG · 2026-02-14

**Summary Of Contributions:**

This paper introduces VaRDASS (Variance-Reduced Domain Adaptation via Stratified Sampling), a method that reduces the variance of stochastic domain discrepancy estimates in unsupervised domain adaptation (UDA). The authors focus on two widely used discrepancy measures  MMD and CORAL and derive tailored stratification objectives for each. For MMD, they show that minimizing the variance of the empirical mean embedding is a valid surrogate for minimizing the variance of the squared MMD estimate, and prove this is theoretically optimal when the covariance is isotropic (Theorem 1). For the anisotropic case, they provide expected and worst-case error bounds (Theorems 2–3) parameterized by Spearman's rank correlation. A similar surrogate is derived for CORAL (Theorem 4). The resulting optimization is a dynamically-weighted kernel k-means problem, for which a greedy, parallelizable algorithm is proposed. Experiments on Camelyon17, Humpbacks, and Spawrious show improved target-domain accuracy when VaRDASS is used with MMD and CORAL losses.

**Key Strengths:**

The paper addresses a real and underappreciated problem: high variance in domain discrepancy estimation can undermine the theoretical benefits of UDA methods. The idea of using stratified sampling specifically tailored to the structure of MMD and CORAL losses is novel and well-motivated. The theoretical development is careful, progressing logically from the isotropic optimality result to practical error bounds for the anisotropic case.

**Key Weaknesses:**

The experimental evaluation, while showing consistent trends, is limited in scale and the improvements are sometimes within standard error margins. The theoretical results rely on assumptions (Gaussian embeddings, isotropic covariance) whose validity in practice is not verified. The method introduces nontrivial hyperparameters (re-clustering frequency TT
T, number of random trials, parallel assignments) whose sensitivity is only partially explored.

**Additional Comments:**

The paper is generally well-written and well-organized. The progression from theory to algorithm to experiments is logical. I appreciate the honesty in the limitations section (Section 4)  the NP-hardness of k-means, scaling concerns, and singleton strata issues are all legitimate. The paper would benefit most from stronger experimental evidence (more repeats, wall-clock times, larger datasets).

**Audience:**

Yes

**Audience Explanation:**

Variance reduction in stochastic optimization is a mature topic, but its application to UDA-specific losses (MMD, CORAL) is genuinely novel. The observation that classical variance reduction methods (SAGA, SVRG) are incompatible with these losses due to their non-finite-sum structure is well-made and identifies a real gap. Practitioners working on domain adaptation  particularly in settings with small datasets or minibatches  would find the approach relevant. The connection between stratified sampling and kernel k-means clustering is elegant and could inspire related work in other areas where pairwise or distributional losses are optimized stochastically.

**Claims And Evidence:**

Yes

**Claims Explanation:**

N/A

**Requested Changes:**

1- Statistical rigor of experimental comparisons. Three repeats is insufficient for reliable conclusions when many comparisons are within standard errors. I would expect at least 5  repeats, or alternatively, a proper statistical test   across dataset splits. Without this, the claim that VaRDASS "performs best overall" is not well-supported for all settings.

2- Clarify computational overhead. The paper mentions O(kns) for Algorithm 2, but the full pipeline includes kernel matrix construction  O(n_s^2)  , kernel k-means iterations, and periodic re-clustering every T steps. A wall-clock comparison with uniform sampling would be essential for practitioners. How much slower is training end-to-end?

3- Larger-scale experiments. All three datasets are relatively small (8k–18k examples). Even a single experiment on a larger benchmark (e.g., Office-Home, DomainNet) would strengthen the practical relevance, especially given the acknowledged quadratic scaling concern.

---

> ### Author Response · Authors · 2026-03-06
>
> Thank you for your feedback.
>
> 1. We have increased the number of experimental repeats to 5. Note that there were also some hyperparameter changes in response to another reviewer, so the new results are not directly comparable with the originals.
> 2. We have added a comparison of wall-clock training times (Table 3).
> 3. We have also added results on the Office-Home dataset.

---

### Review · Reviewer_wR9N · 2026-02-21

**Summary Of Contributions:**

This paper addresses the high-variance issue in distribution-matching frameworks for unsupervised domain adaptation. It proposes optimal stratification objectives for two classical frameworks — correlation alignment (CORAL) and maximum mean discrepancy (MMD) — and provides theoretical justification under certain assumptions. To realize the stratification in practice, the paper introduces a dynamically weighted kernel k-means algorithm.

Strengths

- The proposed stratification method can be integrated as an add-on to existing discrepancy-based frameworks such as MMD and CORAL, without altering the core learning objective.
- The paper provides theoretical analysis showing that minimizing the variance of the empirical mean embedding is equivalent to minimizing the variance of the squared MMD  under isotropy assumptions. This offers a principled foundation for the surrogate objective.
- The dynamically weighted kernel k-means implementation is conceptually simple and reasonably well designed.

Weakness

- A straightforward way to reduce the variance of minibatch estimators is to increase the batch size. The batch size used in this paper is sampled as: $k\sim 2^{\text{U}(3, 5.5)}$. This is relatively small for distribution-matching frameworks. In contrast, prior classical works typically used much larger batch sizes (e.g., CORAL/DANN: 128)
- The paper does not evaluate on widely used UDA benchmarks like Office-31 (31 classes), Office-Home (65 classes) and DomainNet (345 classes). Instead, it evaluates on Camelyon17 (2 classes), Humpbacks (2 classes), and Spawrious (4 classes). These datasets involve very low class cardinality. Therefore, the current evaluation does not convincingly demonstrate applicability to standard multi-class UDA scenarios.
- I was unable to find explicit reference paragraphs in the main text discussing Appendix Tables 3–5. The results appear somewhat inconsistent. It is not straightforward to determine the superiority of the proposed method.

**Audience:**

Yes

**Audience Explanation:**

Distribution-matching frameworks are classical approaches in unsupervised domain adaptation. Addressing the high-variance issue in these frameworks with simple stratification objectives would likely be of interest to part of the community.

**Claims And Evidence:**

No

**Claims Explanation:**

I believe the empirical evidence is not sufficient to justify the claims in more practical settings, as discussed in the weaknesses above.

**Requested Changes:**

- The paper would benefit from experiments using larger batch sizes to better substantiate the central claim regarding variance reduction.
    - In Figure 4, the minibatch size is evaluated only up to $k=40$. It would be informative to extend this analysis to larger batch sizes (e.g., 64, 128, 256) to observe whether the variance gap between different components persist as $k$ increases.
    - Additionally, the main experimental results (Tables 1 and 2) could be reported under more standard batch sizes such as 128. This would allow a clearer comparison with prior discrepancy-based methods (e.g., CORAL, DANN, MMD), which were typically trained with batch sizes around 128–256.
- The current evaluation is conducted on datasets with relatively small numbers of classes. While these datasets are valid, they do not reflect the complexity of standard UDA benchmarks. It would substantially strengthen the paper to include results on at least one widely used multi-class benchmark mentioned above.
- The paper should include paragraphs to explain the tables in the appendix. Any performance inconsistencies should also be discussed.

---

> ### Author Response · Authors · 2026-03-06
>
> Thank you for your feedback.
>
> 1. We have extended Figure 4 to $k=256$, and repeated all the experiments using a new hyperparameter search range of 8-128 for the batch size.
> 2. We have also added results on the Office-Home dataset.
> 3. The tables in Appendix C are now also explicitly referenced in the main text, and have also been reformatted to hopefully make the comparisons clearer.

---

### Decision · Action_Editor_CHqk · 2026-04-01

**Recommendation:** Accept as is

**Audience:**

Yes

**Audience Explanation:**

UDA is a well-studied area, and leveraging stratified sampling to reduce loss variance is a novel and meaningful contribution. The insight that variance can be mitigated through improved sampling, not just architectural changes, is valuable for both theorists and practitioners in the TMLR community.

**Claims And Evidence:**

Yes

**Claims Explanation:**

The paper applies stratified sampling to reduce the variance of minibatch estimates of domain discrepancy losses in unsupervised domain adaptation (UDA). It derives stratification objectives for both MMD and CORAL, provides expected and worst-case error bounds for MMD, and proves that the MMD objective is variance-optimal under certain assumptions. The strata are constructed via a weighted kernel k-means algorithm.

Overall, the claims are well-supported by both theoretical analysis and empirical evidence.

In response to the initial reviews, the authors have made several improvements:
- Increased experimental repeats to 5
- Extended the batch size range (8–128)
- Added results on the Office-Home dataset
- Included comparisons with more baselines (CDAN+ SDAT, CDAN+ELS)
- Reported wall-clock training times
- Added analysis of the effect of \(T\) on performance
- Clarified and explicitly referenced Appendix tables
- Expanded the discussion on post-training discrepancy

All reviewers are satisfied with these revisions and are supportive of acceptance.